# Administration of novobiocin and apomorphine mitigates cholera toxin mediated cellular toxicity: Lessons from cholera toxin yeast model system

Sonali Eknath Bhalerao[1], Himanshu Sen[1,2], Saumya Raychaudhuri[1,2]*

**1** CSIR-Institute of Microbial Technology, Chandigarh, India, **2** Academy of Scientific and Innovative Research (AcSIR), Ghaziabad, India

* saumya@rocketmail.com, saumya@imtech.res.in

**Data Availability Statement:** All relevant data are within the article and its Supporting Information files.

## Abstract

Cholera is a dreadful disease. The scourge of this deadly disease is still evident in the developing world. Though several therapeutic strategies are in practice to combat and contain the disease, there is still a need for new drugs to control the disease safely and effectively. Keeping in view the concern, we first successfully established an inducible yeast model to express cholera toxin subunit A, and then used this yeast model, to screen a small molecule library against cholera toxin A subunit. Our effort resulted in the discovery of a small molecule, apomorphine (a Parkinson's disease drug) effective in reducing the lethality of toxic subunit in yeast model. In addition, novobiocin, an inhibitor of ADP ribosylation process, a key biochemical event through which cholera toxin exerts its action on host, was also found to rescue yeast cells from cholera toxin A subunit mediated toxicity. Finally, the effects of both molecules were tested on the cholera toxin-treated human gut epithelial cell line HT29, and it was observed that both apomorphine and novobiocin prevented cholera toxin-mediated cellular toxicity on HT29 intestinal epithelial cells.

## Introduction

*Vibrio cholerae* causes cholera, a life-threatening diarrheal disease with worldwide distribution [1]. Out of 220 serogroups, strains belonging to serogroups O1 and O139 are associated with epidemic cholera. Though decades of continuing efforts have harvested wealth of information on the biology of the bacterium and disease per se, cholera continues to rage and remains a major public health problem in the developing and war-torn countries. As per WHO records, over 3 million people are affected annually, leading to over 100,000 deaths [2]. Among multiple lines of treatments, oral or intravenous rehydration remains the mainstay in cholera. Antibiotics are also given as an adjunct therapy to control and decrease duration of the disease. But indiscriminate use of antibiotic increases in the emergence of multidrug resistant strains of *V. cholerae* [3,4], thereby contributing towards global burden of antimicrobial resistance (AMR). Vaccines with limited efficacy providing short term protection are also available and can be

**Funding:** This work was partly supported by grants from CSIR-IMTECH (OLP 151), CSIR-MLP39, and the Science and Engineering Research Board (CRG/2018/000297/SERB-GAP/0185). Sonali E. Bhalerao and Himanshu Sen acknowledge the Council of Scientific and Industrial Research (CSIR) for their fellowships. Sonali E. Bhalerao also acknowledges support from CSIR-MLP39.

**Competing interests:** NO authors have competing interests

useful to restrict the spread of cholera in outbreak prone area. But the vaccines can give only 65% protection against cholera for 6 months up to 3 years [5]. Therefore, there is a desperate need to come up with newer strategies to combat cholera.

With continuing effort from researchers across the globe, new areas are increasingly emerging to tackle the pathogenesis and fitness of the organism. So far, this is achieved either by chemically synthesized small molecules jeopardizing the function of regulatory proteins (e.g ToxT, LuxO) involved in the production of virulence factors [6–12] or gut commensal derived metabolites targeting the overall fitness and survival of the organism in the host intestine [13–20]. Though these strategies hold a promise in controlling the disease, the safety and effectiveness of each strategy under clinical settings yet to be evaluated. Interestingly, natural variants of regulatory proteins are also identified and found resistant to small molecule mediated functional modulation [21]. Probiotic mediated septicemia is also evidenced in animal model as well as in ICU patients and raising grave concern over the usage of probiotic strains [22,23]. Recently, L-ascorbic acid is shown to restrict the growth of *Vibrio cholerae* under *in vitro* conditions [24].

Keeping view of all recent advancements and issues associated with different strategies to combat cholera, we were driven by a desire to explore further in the same direction with a hope to come up with better molecule for abating the disease.

*Saccharomyces cerevisiae*, the budding yeast has been exploited as a non-mammalian model system for functional evaluation of several virulence factors from many pathogenic bacteria including *V. cholerae* [25–32]. Simple ectopic expression of virulence factors can lead to a spectrum of discernible phenotypes in yeast that aid to build testable hypotheses regarding their function and roles in pathogenesis. Not only advancing functional understanding of microbial and higher eukaryotic proteins, such yeast model systems are also instrumental in the discovery of novel small molecule modulators against specific proteins further prompted us to embark on the present work [33,34].

In this work, we developed a yeast model of cholera toxin by ectopically expressing gene encoding cholera toxin A (CTXA) subunit. The expression of CTXA resulted a severe growth defect in the recombinant yeast. Subsequently, the CTXA expressing yeast was subjected to screen against small molecule library and one small molecule, apomorphine, was found to inhibit CTXA mediated growth retardation. In addition, we also observed another small molecule novobiocin, a known inhibitor of ADP ribosylation also reduced CTXA toxicity in recombinant yeast strain.

Cholera toxin is demonstrated to alter cellular morphology in PC12 cells [35]. We also documented morphological changes of HT29 intestinal cell lines mediated by commercially available cholera toxin (CT) and such cellular alterations was also reduced by exogenous administration of apomorphine and novobiocin. Furthermore, this study also reveals the effectiveness of apomorphine, a dopamine agonist commonly used to treat Parkinson disease, in controlling cholera toxin mediated cellular toxicity and also demonstrates the strength of yeast model of virulence factors in drug discovery.

## Materials and methods

### Yeast strains and culture conditions

The *S. cerevisiae* strains BY4741 used in this study (S1 Table) were grown in YPD (1% yeast extract, 2% peptone, 2% glucose) broth or agar plates (2%) at 30˚C. Yeast strains harboring plasmids were cultured in appropriate selective SC (synthetic complete) liquid media (0.67% Yeast nitrogen base without amino acids, 2% glucose, addition appropriate amino acids and nucleotide bases) or agar plates (2%). For 100 ml media, the concentration of amino acids and

bases is used as follows: methionine 4 mg, histidine 4 mg, leucine 6 mg, tryptophan 4 mg, adenine 4 mg and uracil 4 mg. In case of SC$^{Raf}$ and SC$^{Gal}$ media, glucose is replaced with raffinose and galactose respectively. Yeast strains transformed with KanMX4 selection marker were selected by plating them on YPD agar plates containing G418 (200 μg/mL).

### Cell line and culture conditions

Human intestinal epithelial cell line (HT29) (ATCC HTB-38) (S1 Table) was cultured in RPMI 1640 media with 10% heat inactivated FBS and 1X antibiotic-antimycotic. Cell cultures were incubated at 37°C, 5% $CO_2$.

### Gene cloning

Gene for cholera toxin A subunit without signal sequence was cloned under a GAL10 promoter into pESC-Leu, pGML10 vectors and HO-locus of the yeast strain of BY4741. The primers used for the cloning of *ctxA* are listed along with their sequences in S2 Table.

### Yeast growth assay

The desired plasmid constructs were transformed into *S. cerevisiae* strains. The transformed cells were selected by plating on selective SC solid agar plates. Three colonies of each clone were inoculated into selective SC$^{Raf}$ media and grown overnight at 30°C. The overnight grown cultures were used to set up secondary cultures in fresh media with a starting $A_{600}$ of 0.1, the cultures were allowed to grow till mid log phase. The effect of cholera toxin on yeast was examined by spotting equal number of cells on SC and SC$^{Gal}$ plates lacking corresponding auxotrophic markers to maintain the plasmids. Yeast growth was monitored for 60–70 h and images were documented. For liquid growth assay, the cultures grown in SC$^{raf}$ media were diluted to $A_{600}$ 0.05 in 25 ml of induction media (galactose) with appropriate nutritional supplementation. The absorbance was measured at 600 nm for stipulated period.

### Small molecule library screening

Primary screening was done in 96 well plates using yeast clones BY4741/pGML10 and BY4741/ctxA no signal/pGML10. Spectrum collection small molecule library (MicroSource Discovery Systems Inc., Gaylordsville) was used to screen molecules against cholera toxin in yeast model. Overnight grown cultures were used to set up secondary cultures of BY4741/pGML10 and BY4741/ctxA no signal/pGML10 at $A_{600}$ nm = 0.15 in YNB-Glucose-HMU media. When culture OD600 nm reached to 0.3–0.4, cultures were induced with 2% galactose and were further incubated at 30°C for 4 h on shaking. After 4 h, ampicillin (50 μM) was added to the cultures to avoid bacterial contamination, and cultures were distributed in 96 well plates (160 μl/well). 20 μM of compounds were also added in 96 well plates from small molecule library. Using multichannel pipette, cultures and compounds were mixed well and plated were incubated for 24 h at 30°C on static. The effect of drug molecules interfering toxin action and subsequent growth restoration was examined by spotting the cultures on SC and SC$^{Gal}$ plates using prong replicator. Yeast growth was monitored for 60–70 h and images were documented.

Tertiary screening was done using an overnight grown cultures of BY4741/pGML10 and BY4741/ctxA no signal/pGML10 to set up a secondary culture at starting $A_{600}$ nm = 0.1 in YNB-glucose-HMU media. These cultures were incubated at 30°C on shaking till OD600 nm reached to 0.5 for induction with 2% galactose. Different concentrations of small molecule compounds ranging from 10 μM to 1 mM were also added at the same time. The effect of drug

molecules interfering toxin action and subsequent growth restoration was examined by spotting the cultures on SC and SC$^{Gal}$ plates after 6 h incubation at 30˚C on shaking. Yeast growth was monitored for 60–70 h and images were documented.

## Cell line morphology study

HT29 cells were cultured in 12 well plate to 60% confluence. Media was discarded and cells were washed 2X with 1X DPBS and fresh media was added to the wells. Cholera toxin obtained from sigma (10 ng/ml and 40 ng/ml) and drug molecules (apomorphine- 10 μM, 20 μM and novobiocin- 100 μM, 200 μM, Apo 10 μM + Novo 100 μM) were added to the wells and mixed properly by gentle shaking. Plates were kept at 37˚C, 5% CO$_2$ for incubation. After 6 h, pictures were taken under light microscope at 20X resolution.

## Sample preparation for flow cytometry

HT29 cells were grown to 50% confluence in 12 well plate and were treated with CT (10 ng/ml and 40 ng/ml) and drug molecules (apomorphine– 10 μM, 20 μM and novobiocin– 100 μM, 200 μM, Apo 10 μM + Novo 100 μM). After 24 h of treatment, media were collected and centrifuged at 850 g for 10 mins to collect dead cells floating in the media. This was followed by trypsin treatment and cells were collected through centrifugation at 850 g for 5 mins. Positive controls were prepared by treating cells at 94˚C for 3 mins. Cells were kept on ice for further treatment. 200 μl, propidium iodide stain (25 μg/ml) and 0.3 μl LIVE/DEAD™ fixable green dead cell stain (Thermofisher scientific) was added to the culture pellets, mixed properly and were kept on ice for 10 mins. After staining, cells were washed with 1X DPBS and were resuspended in 500 μl 1X DPBS. Samples were sorted immediately with BD FACSVerse cell analyzer using 2 different excitation emission spectrums of wavelengths 488 nm and 566 nm. Data were analyzed using FlowJo software. The live cell percentage and mean fluorescence intensity for the LIVE/DEAD™ Fixable Green Dead Cell Stain (566 nm) were plotted in MS Excel to compare membrane permeability between drug-treated and toxin-treated cells.

## Results

### Ectopic expression of cholera toxin A subunit in *Saccharomyces cerevisiae* affects cell growth

The gene fragment containing cholera toxin A subunit (*ctx*A) without secretion signal was amplified and cloned under the control of GAL-1 promoter in high copy number vector pESC Leu to generate recombinant plasmid pESC-Leu-CTXA. To investigate the toxicity of CTXA, the recombinant plasmid pESC-Leu-CTXA was transformed into *S. cerevisiae* BY4741. Under inducing condition in the presence of galactose, culture spotting on solid agar media exhibited strong growth inhibition of the host *S. cerevisiae* expressing CTXA, whereas no effect on growth was observed in the host expressing only vector (Fig 1A). There was also weak growth inhibition under repressive (glucose) condition. This is due to leaky expression from GAL1 promoter of pESCLeu. The leaky expression issue related to GAL1/10 promoter has been noticed in previous studies as well [31]. The growth inhibition was recapitulated in the liquid growth assay where the difference in the OD 600nm of control and CTXA expressing yeast was clearly visible (Fig 1B). It is documented that lethality of some effector proteins is also linked with a high level of expression [36]. In some cases, higher expression also leads to nonspecific toxic effects. To ascertain, low levels of expression of CTXA maintains similar toxicity, the gene was cloned in pGML10, a low copy number vector (S1 Table). *S. cerevisiae* strain BY4741 transformed with pGML10-CTXA exhibited lethal phenotype in inducing condition

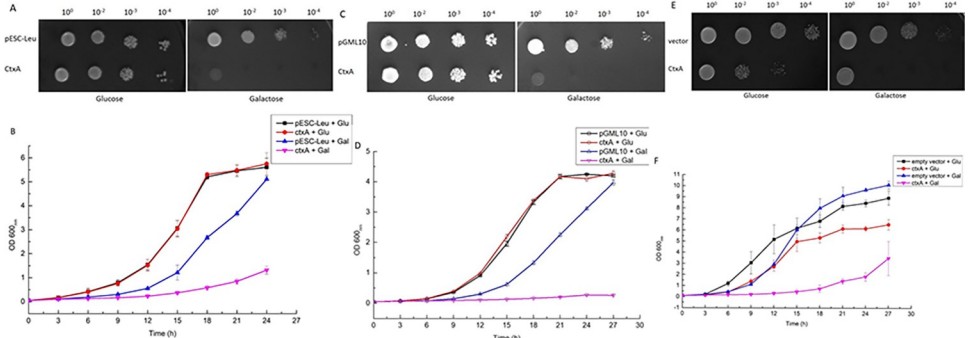

**Fig 1.** Effect of *ctxA* expression on yeast growth: A and B. Effect of *ctxA* expression on yeast growth when expressed in high copy number vector: *ctxA* was cloned in a high copy number vector pESC-Leu and was expressed in BY4741 strain of yeast. For liquid growth assay, the secondary cultures were set up at a starting $OD_{600}$ 0.05 and were monitored for 24 h. $OD_{600}$ was checked every 3 h. The data shown here is collected from 3 independent experiments. For spotting data, secondary cultures were set up at starting $OD_{600}$ 0.1, after 6 h, cultures were serially diluted ($10^0$, $10^2$, $10^3$, $10^4$) and were spotted on glucose and galactose plates. Pictures were taken after 60–70 h. C and D. Effect of *ctxA* expression on yeast growth when expressed in a low copy number vector: *ctxA* was cloned in a low copy number vector pGML10 and was expressed in BY4741 strain of yeast. Liquid growth assay and spotting assay was performed for 27 h as mentioned in A. E and F. Effect of *ctxA* expression on yeast growth when single copy of gene is expressed: *ctxA* was expressed in a chromosomal DNA of BY4741 using HO-pGAL-polykanMX4-HO vector. Liquid growth was performed for 27 h as explained above. For spotting experiment, the secondary cultures were grown to $OD_{600}$ 0.8–0.9 and spotted on solid agar containing glucose and galactose.

(Fig 1C and 1D). In addition, gene encoding CTXA was also chromosomally integrated at HO-locus. Upon induction with galactose, the recombinant strain (BY4741/HOΔ::HO-CTXA-kanMX4-HO) exhibited CTXA mediated lethality (Fig 1E and 1F). It should be noted that remaining experiments were carried out in this recombinant strain harboring chromosomally integrated singly copy of gene encoding CTXA.

## A small molecule apomorphine mitigated CTXA mediated growth retardation in recombinant yeast model system: Lessons from phenotypic screen

Thus far, our results suggested that expression of cholera toxin A subunit is detrimental for yeast growth. Therefore, any molecule perturbing the lethality caused by CTXA should restore the growth of our recombinant yeast strain. Keeping this in mind, we engaged on a phenotypic screening assay to find small molecule modulator (s) of cholera toxin. In pursuit of our interest, we subjected our recombinant *S. cerevisiae* strain harboring *ctxA* gene (S1 Table) to screen 2560 small molecules using the Spectrum collection small molecule library. This unique library of small molecules possesses biologically active and structurally diverse compounds. The thirty- two 96 well plates of the small molecule library screening eventually resulted one positive hit which appeared to rescue the viability of the yeast cells expressing CTXA (Fig 2A and 2B). The molecule is apomorphine, a dopamine agonist [37]. To evaluate further the efficacy of apomorphine in growth restoration of the recombinant yeast, liquid growth assay coupled with viability spotting was done with various concentrations of the drug ranging from 20 μM to 200 μM. We observed that apomorphine showed better rescue of the recombinant strain if added twice in the cultures during the entire span of liquid growth assay (Fig 2C–2E). Taken together, our study clearly demonstrated the utility of yeast model system in facilitating discovery of novel inhibitor against cholera toxin and the potential of apomorphine to be tested further against cholera toxin.

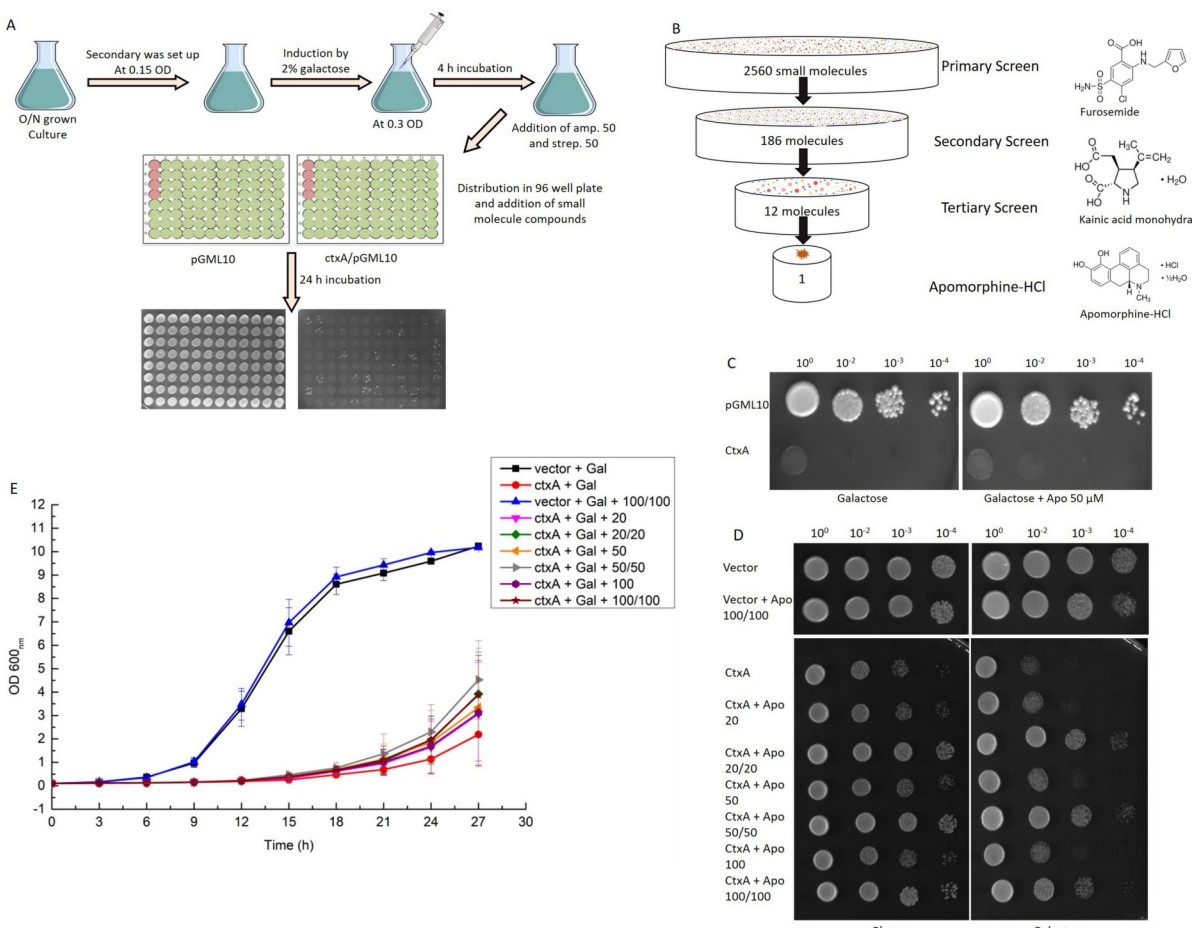

**Fig 2. Small molecule library screening using cholera toxin yeast model system.** A. Schematic of high-through put screening: Pre-induced cultures of BY4741/*ctxA*/pGML10 and BY4741/pGML10 as a control were used. These cultures were evenly distributed in 96 well plates, then compounds from the library plate were added at the final concentrations of 20 μM. Both plates were incubated for 24 h at 30°C and were then spotted on glucose and galactose plates for readouts. B. Schematic representation of the outcome of the small molecule library screening. Structures of the few positive hits from secondary screen are shown. C. Tertiary screening of Apomorphine HCl: Pre-induced cultures of BY4741/*ctxA*/pGML10 and BY4741/pGML10 as a control were spotted on solid agar plate containing galactose and galactose with apomophine (50 μM). Pictures were taken after 60–70 h D and E. Effect of Apomorphine HCl on cholera toxicity in yeast: Secondary cultures of BY4741/*ctxA*-HO and vector control were set up at starting $OD_{600}$ 0.05 in glucose and galactose media with different concentrations of apomorphine and were monitored for 27 hrs. $OD_{600}$ was checked every 3 h. The data shown here is collected from 3 biological triplicates (D). At the end point of liquid growth assay, the cultures were serially diluted ($10^0$, $10^2$, $10^3$, $10^4$) and were spotted on glucose and galactose plates. Pictures were taken after 60–70 hrs (E). Liquid growth assay.

## Novobiocin, an ADP ribosylation inhibitor rescued recombinant yeast from cholera toxin mediated lethality

Mechanistically cholera toxin, more specifically the enzymatic A subunit of cholera toxin, activates Gsα by irreversible transfer of ADP moiety of $NAD^+$ to $Arg^{201}$ of Gsα [38–40]. The ADP-ribosylation of G protein triggers cascade of events involving hyper activation of cellular adenylate cyclase, followed by a sharp rise in intracellular level of cyclic AMP. The increase c-AMP leads to uncontrolled secretion of fluid and electrolytes into the lumen of the small intestine [41–43]. Assuming CTXA exerts its lethality in our yeast model through ADP ribosylation, therefore, blocking ADP ribosylation should lessen the lethality of cholera toxin. Keeping this in mind, we selected novobiocin, a known mono ADP ribosylation inhibitor [44] and

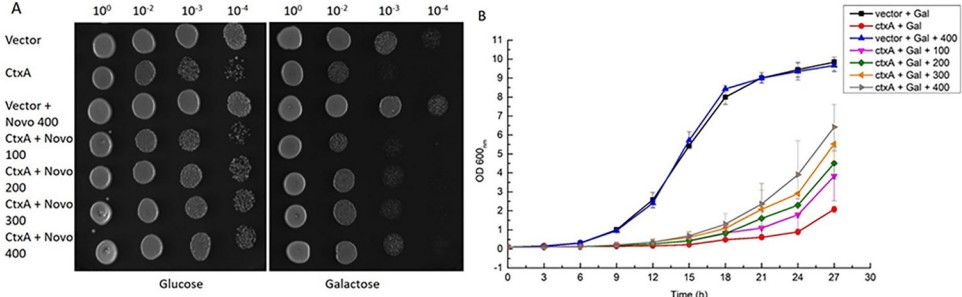

**Fig 3. Effect of novobiocin on cholera toxicity in yeast: Secondary cultures of BY4741/ctxA-HO and vector control were set up at starting OD$_{600}$ 0.05 in glucose and galactose media with different concentrations of novobiocin and were monitored for 27 h.** OD$_{600}$ was checked every 3 h. The data shown here is collected from 3 independent experiments (A). At the end point of liquid growth assay, the cultures were serially diluted ($10^0$, $10^1$, $10^2$, $10^3$) and were spotted on glucose and galactose plates. Pictures were taken after 60–70 h (B). Liquid growth assay.

examined its effect on the reversal of toxin mediated toxicity in the recombinant yeast. We observed that novobiocin reduced the toxicity caused by the CTXA in the yeast similar to apomorphine in both the culture viability spotting and liquid growth assay (Fig 3A and 3B).

## Apomorphine and novobiocin restricted CT mediated cellular toxicity and promoted survival of intestinal cells

Cholera toxin (CT) causes morphological changes of intestinal epithelial cells [35], therefore, we desired to investigate whether CT mediated morphological changes and cellular death can be restricted by the exogenous administration of apomorphine and novobiocin. Towards this end, we treated HT29, a well characterized intestinal epithelial cell line with cholera toxin (Sigma) with various concentrations of cholera toxin and observed a dose dependent morphological changes of HT29 cells (Fig 4A). It should be noted that the HT29 cell line has been extensively exploited in cholera research [45–47]. To gain further insight on the viability status of epithelial cells, a FACS based live-dead assay was performed. Survival of intestinal cells was reduced approximately to 50–55% in the presence of cholera toxin after 24 h of treatment as measured by flow cytometry analysis (Fig 4B). To examine whether apomorphine and novobiocin can reverse cellular death caused by cholera toxin, various concentrations of drugs were added simultaneously with different concentrations cholera toxin on HT29 cells. After stipulated period of incubation, cellular morphology was microscopically examined. We observed both apomorphine and novobiocin effectively blocked CT mediated morphological changes and maintained wild type morphology of HT29 cells (Fig 4A). We also observed the rescue of toxin treated cells in the presence of drugs (Fig 4B). Control sets containing drugs either alone or together did not affect the health of intestinal cells at the highest concentration (S1 Fig).

## Discussion

Cholera is still one of the major killer diseases in the developing and war-torn countries. As summarized in the introductory section, there exists a demand towards the development of newer strategies balancing therapeutic safety and efficacy to control the disease in a better way. Keeping this in mind, we embarked on the present program to construct a yeast model of cholera toxin and subject this model to screen a chemical library. We choose yeast model primarily because of our prior engagement with yeast model of type III effectors [29,31,32] and yeast models have been demonstrated highly effective and instrumental in screening small molecule libraries [33,34]. The library "Spectrum collection" from Microsource Discovery systems Inc.

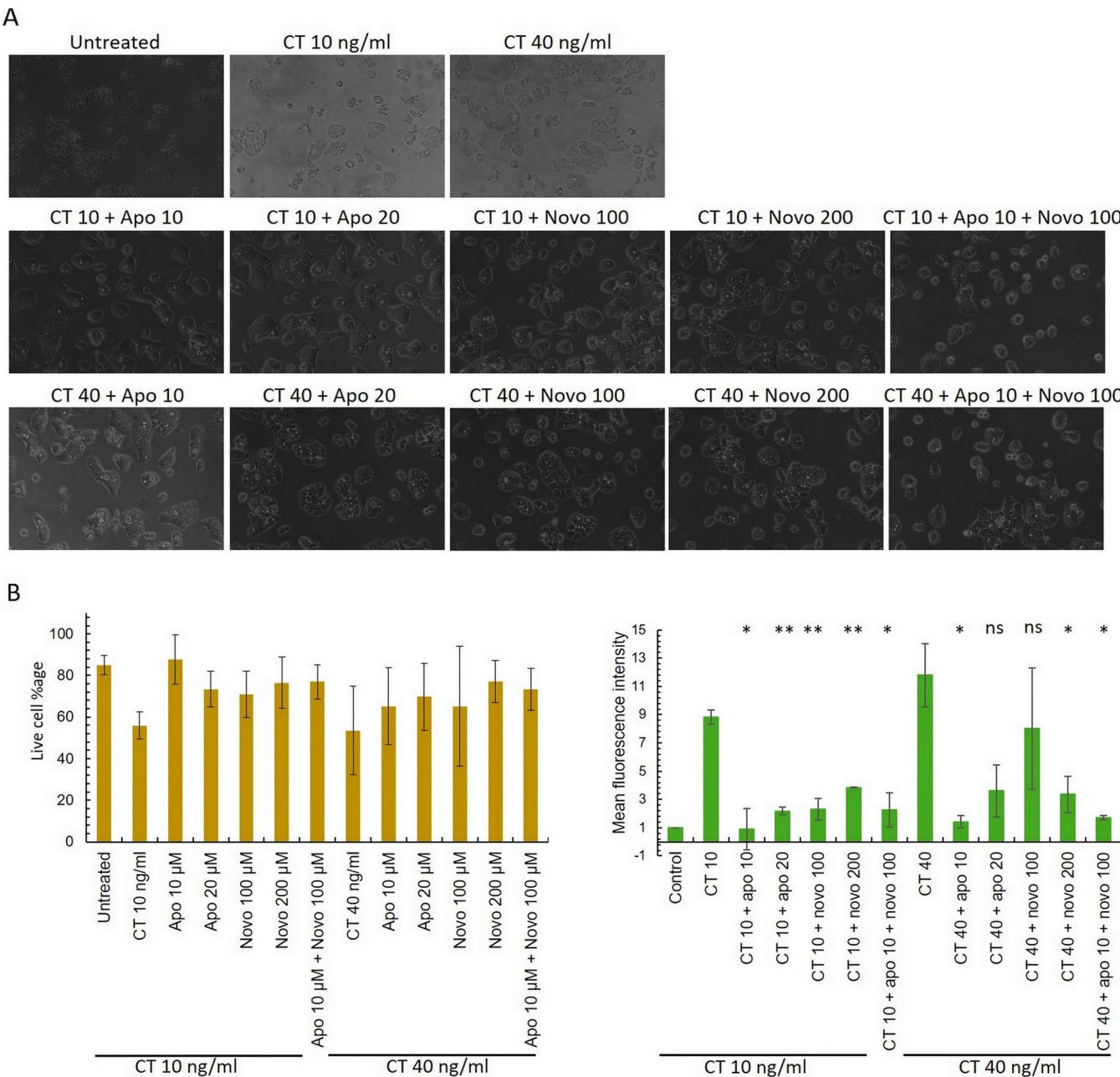

**Fig 4. Effect of apomorphine HCl and novobiocin on CT toxicity on HT29.** A. 0.2 million HT29 cells were seeded in 24 well plates. At 60% confluence, cells were washed 2X with DPBS. Cholera toxin and drugs treatment were given for 6 h. Images were taken at 20X magnification under a light microscope. B. Flow cytometry analysis: Left panel shows the comparison between cell viability after 24 h of toxin and drug treatments and the right panel shows the mean fluorescence intensity of the same samples. The data shown here is collected from 2 independent experiments. Statistical analysis was performed using student's T test (* P<0.05, ** P<0.01, NS- not significant).

used in our study is also exploited by other groups extensively for the discovery of many novel and repurposing drug molecules some of which are now under clinical trials [48]. Keeping view on various successful repurposing drug leads obtained from this library, we started our screening program with this library. Drug repurposing has several advantages over conventional drug discovery programs in terms of safety and product development. Various different

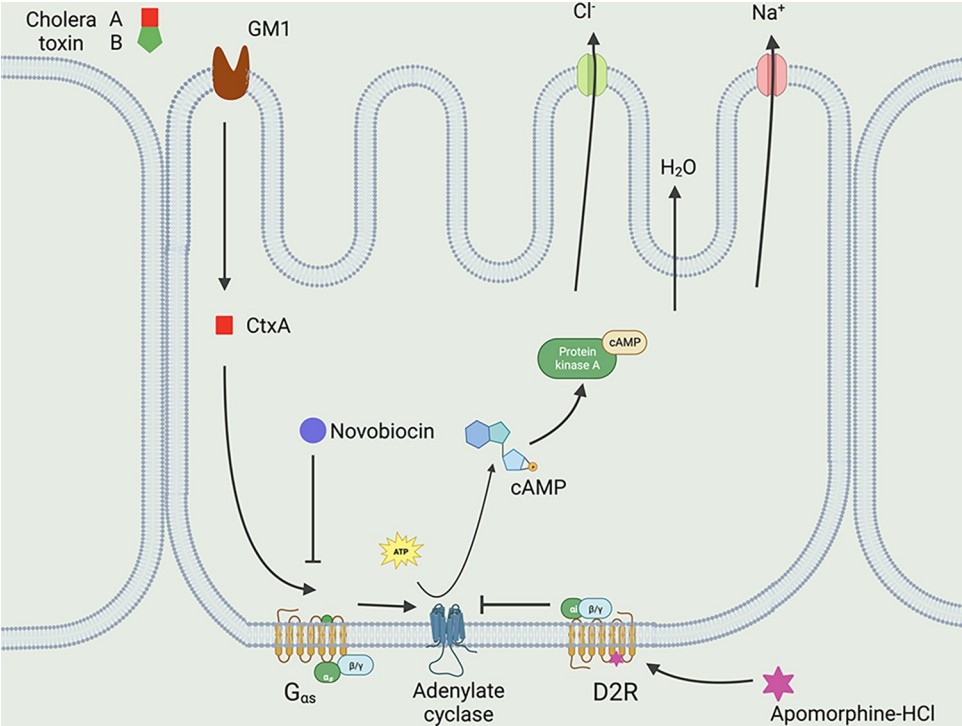

**Fig 5. Predicted mechanism of action of apomorphine HCl and novobiocin against cholera toxin action.**
Schematic was drawn using BioRender and was adapted from Rahman et al., 2018 [56].

methods are used to find out new applications of drug molecules that are different from its original course of action includes *in silico* techniques, literature survey and screening drug libraries. Armed with these strategies, our combined effort on literature survey and screening small molecule library eventually aid in the discovery two molecules namely novobiocin and apomorphine which are inhibiting cholera toxin mediated cellular lethality.

Novobiocin is an aminocoumarine antibiotic produced by actinomycete *Streptomyces niveus* and used to treat MRSA (methicillin resistant *Staphylococcus aureus*) infection. Novobiocin targets DNA gyrase which is a bacterial enzyme that functions as a catalyst for ATP dependent negative supercoiling of closed circular double stranded DNA [49,50]. The molecule also blocks HSP90 (heat shock protein of 90 kDa) chaperone protein in eukaryotic cells. This newly identified function of novobiocin is being explored as a possible anticancer agent [51]. Novobiocin also is an arginine dependent ADP-ribosyl transferase (ART) inhibitor compound. It has been widely used in studies of ARTs to determine their biological functions. This function of novobiocin was exploited in our present study. We observed that novobiocin is able to counter cholera toxin mediated toxicity in our experimental model systems. Considering its function as an ART inhibitor, it is conceivable that novobiocin may interfere with toxin mediated massive production of cAMP, thereby, mitigating toxin mediated cellular lethality (Fig 5).

Apomorphine HCl is clinically used to treat Parkinson's disease. The molecule is also explored as a possible treatment for Alzheimer's disease [52,53]. It is a known dopamine D2 receptor agonist [54]. There are five types of dopamine receptors (D1 to D5), and each receptor has a different function. These five receptors are further divided into two subcategories: D1 and D5 are associated with Gs alpha subunit of G protein coupled receptor (GPCR) and D2, D3, D4 are coupled with Gi subunit of GPCR. Dopamine function in brain depends on the

receptor that it binds to e.g., if it binds to D1R, it will lead to activation of adenylate cyclase through Gs alpha subunit and if it binds to D2R it will inhibit the enzyme through Gi subunit of GPCR [55]. It also has been previously reported by Brann and Jelsema in 1985, that cholera toxin reduces the affinity of apomorphine for dopamine receptors.

Being a D2 receptor agonist, we surmised that binding of apomorphine to its cognate receptor results in the inhibition of adenylate cyclase, thereby perturbing cholera toxin-mediated production of cAMP. Based on our observation, a molecular road map of drug and toxin interaction is shown as a schematic diagram (Fig 5).

G-protein and MAP kinase signaling has been elegantly established in yeast [57]. It would be interesting to examine whether i) the expression of CTXA modulates the G protein and MAP kinase signaling pathways in yeast and ii) whether novobiocin and apomorphine also target the same pathway to counter CTXA mediated effect in yeast. This warrants further investigation.

Recently, new strategies (probiotics and L-ascorbic acid) have emerged to combat cholera targeting the survival and pathogenesis of the cholerae bacterium [16–18,24]. It would be interesting to examine the efficacy of a combination therapy of apomorphine and novobiocin with probiotic or L-ascorbic acid to control the disease much better way.

It is noteworthy to mention that some Parkinson disease drugs affect intestinal physiology and microbiota community structure in healthy rats [58]. Therefore, it is necessary to examine the effect of apormorphine and novobiocin on gut microbiota. Presently, different types of *ex vivo* microbiota fermentation models have been exploited to test the effect of various small molecules and probiotic strains on the gut community composition [59–61]. Additional studies are necessary to address these issues.

## Supporting information

**S1 Fig. Effect of apomorphine HCl and novobiocin on HT29: A.** 0.2 million HT29 cells were seeded per well. At 60% confluence, cells were washed 2X with DPBS. Drugs treatment was given for 6 hrs. Images were taken at 20X magnification under a light microscope. B. Flow cytometry analysis: Comparison between cell viability after 24 hrs of drugs treatment. The data shown here is collected from two independent experiments.
(TIF)

**S1 Table. Strains and plasmids used in the study.**
(DOC)

**S2 Table. Primers used in the study.**
(DOCX)

**S1 Data set.**
(ZIP)

## Acknowledgments

We gratefully acknowledge Dr. Chetna Dureja for providing critical insights in this work. We also acknowledge Dr. Deepak Sharma, Dr. Ravi Mishra and Dr. Ashwani Kumar for providing small molecule library and cell culture facility.

## Author Contributions

**Conceptualization:** Saumya Raychaudhuri.

**Data curation:** Sonali Eknath Bhalerao, Himanshu Sen.

**Formal analysis:** Sonali Eknath Bhalerao, Himanshu Sen.

**Funding acquisition:** Saumya Raychaudhuri.

**Methodology:** Sonali Eknath Bhalerao, Himanshu Sen.

**Project administration:** Saumya Raychaudhuri.

**Resources:** Sonali Eknath Bhalerao, Himanshu Sen.

**Supervision:** Saumya Raychaudhuri.

**Validation:** Himanshu Sen.

**Visualization:** Sonali Eknath Bhalerao.

**Writing – original draft:** Saumya Raychaudhuri.

**Writing – review & editing:** Himanshu Sen, Saumya Raychaudhuri.

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
