## [Decision Letter · Decision Letter 0]

15 Oct 2024

PONE-D-24-21529Administration of novobiocin and apomorphine mitigates cholera toxin mediated cellular toxicity: Lessons from cholera toxin yeast model systemPLOS ONE

Dear Dr. Raychaudhuri,

Thank you for submitting your manuscript to PLOS ONE. After careful consideration, we feel that it has merit but does not fully meet PLOS ONE’s publication criteria as it currently stands. Therefore, we invite you to submit a revised version of the manuscript that addresses the points raised during the review process.

We look forward to receiving your revised manuscript.

Kind regards,

Rui Tada, Ph.D.

Academic Editor

PLOS ONE

Journal Requirements:

We gratefully acknowledge Dr. Chetna Dureja for providing critical insights in this work. We

also acknowledge Dr. Deepak Sharma, Dr. Ravi Mishra and Dr. Ashwani Kumar for

providing small molecule library and cell culture facility. This work was partly supported by

grants from CSIR-IMTECH (OLP 151), CSIR-MLP39 and Science and Engineering

Research Board (CRG/2018/000297/SERB-GAP/0185). Sonali E Bhalerao and Himanshu

Sen, acknowledge Council of Scientific and Industrial Research (CSIR) for fellowships.

Sonali E Bhalerao also acknowledges CSIR-MLP39.

4. We note that your Data Availability Statement is currently as follows: All data are included in the manuscript.

Reviewers' comments:

Reviewer's Responses to Questions

**Comments to the Author**

1. Is the manuscript technically sound, and do the data support the conclusions?

Reviewer #1: Yes

Reviewer #2: Yes

Reviewer #3: Yes

2. Has the statistical analysis been performed appropriately and rigorously? 

Reviewer #1: No

Reviewer #2: Yes

Reviewer #3: Yes

3. Have the authors made all data underlying the findings in their manuscript fully available?

Reviewer #1: Yes

Reviewer #2: Yes

Reviewer #3: Yes

4. Is the manuscript presented in an intelligible fashion and written in standard English?

Reviewer #1: Yes

Reviewer #2: Yes

Reviewer #3: Yes

5. Review Comments to the Author

Reviewer #1: The manuscript presents a study on the mitigation of cholera toxin (CTX) mediated cellular toxicity using novobiocin and apomorphine. Employing a yeast model system and subsequent validation in HT29 intestinal epithelial cells, the authors explore the efficacy of these compounds in reducing CTX-induced toxicity. The study is innovative, leveraging a high-throughput screening approach to identify potential therapeutic agents.

Strengths of the study:

• Innovative Use of Yeast Model: The use of S. cerevisiae as a model organism for screening small molecules is innovative and provides a high-throughput method for identifying potential therapeutic agents.

• Robust Screening Process: The screening of a diverse small molecule library led to the identification of promising compounds, showcasing a systematic approach to drug discovery.

• Relevance to Public Health: The study addresses a critical need for new cholera treatments, which could have significant implications for global health, particularly in cholera-endemic regions.

Limitations:

1. The study is limited to in vitro models, specifically yeast and HT29 cell lines. While these models are valuable for preliminary screening, the lack of in vivo data is a significant limitation. Animal studies are necessary to confirm the efficacy and safety of novobiocin and apomorphine in living organisms.

2. The discussion section should include a comparison of the efficacy and potential advantages of novobiocin and apomorphine over existing cholera treatments and other emerging therapeutic approaches. This context would provide a clearer picture of the significance of the findings.

3. Statistical significance analysis must be performed for all the graphs in the figures.

The manuscript is recommended for publication after addressing the limitations listed above.

Reviewer #2: Dear authors,

It is a good work that i think it contribute to the biological and toxicological sciences. I have one comment to improve your work as follows:

Please, prepare a more rigor and professional abstract showcasing your methodologies and results, and the contributions to the science.

Reviewer #3: The resolution of figures seem very low. I could not figure out some of the labeling. I would increase the resolution and size accordingly. I was unable to verify the first two diagrams because of this issue.

6. PLOS authors have the option to publish the peer review history of their article (what does this mean?). If published, this will include your full peer review and any attached files.

Reviewer #1: No

Reviewer #2: **Yes: **Mohammad Amrollahi-Sharifabadi, Ph.D.

Reviewer #3: No

---

## [Author Response · Author response to Decision Letter 0]

11 Nov 2024

PONE-D-24-21529

Administration of novobiocin and apomorphine mitigates cholera toxin mediated cellular toxicity: Lessons from cholera toxin yeast model system

PLOS ONE

Dear Dr. Raychaudhuri,

Thank you for submitting your manuscript to PLOS ONE. After careful consideration, we feel that it has merit but does not fully meet PLOS ONE’s publication criteria as it currently stands. Therefore, we invite you to submit a revised version of the manuscript that addresses the points raised during the review process.

We look forward to receiving your revised manuscript.

Kind regards,

Rui Tada, Ph.D.

Academic Editor

PLOS ONE

Journal Requirements:

We gratefully acknowledge Dr. Chetna Dureja for providing critical insights in this work. We

also acknowledge Dr. Deepak Sharma, Dr. Ravi Mishra and Dr. Ashwani Kumar for

providing small molecule library and cell culture facility. This work was partly supported by

grants from CSIR-IMTECH (OLP 151), CSIR-MLP39 and Science and Engineering

Research Board (CRG/2018/000297/SERB-GAP/0185). Sonali E Bhalerao and Himanshu

Sen, acknowledge Council of Scientific and Industrial Research (CSIR) for fellowships.

Sonali E Bhalerao also acknowledges CSIR-MLP39.

4. We note that your Data Availability Statement is currently as follows: All data are included in the manuscript.

Reviewers' comments:

Reviewer's Responses to Questions

Comments to the Author

1. Is the manuscript technically sound, and do the data support the conclusions?

Reviewer #1: Yes

Reviewer #2: Yes

Reviewer #3: Yes

2. Has the statistical analysis been performed appropriately and rigorously?

Reviewer #1: No

Reviewer #2: Yes

Reviewer #3: Yes

3. Have the authors made all data underlying the findings in their manuscript fully available?

Reviewer #1: Yes

Reviewer #2: Yes

Reviewer #3: Yes

4. Is the manuscript presented in an intelligible fashion and written in standard English?

Reviewer #1: Yes

Reviewer #2: Yes

Reviewer #3: Yes

5. Review Comments to the Author

Reviewer #1: The manuscript presents a study on the mitigation of cholera toxin (CTX) mediated cellular toxicity using novobiocin and apomorphine. Employing a yeast model system and subsequent validation in HT29 intestinal epithelial cells, the authors explore the efficacy of these compounds in reducing CTX-induced toxicity. The study is innovative, leveraging a high-throughput screening approach to identify potential therapeutic agents.

Strengths of the study:

• Innovative Use of Yeast Model: The use of S. cerevisiae as a model organism for screening small molecules is innovative and provides a high-throughput method for identifying potential therapeutic agents.

• Robust Screening Process: The screening of a diverse small molecule library led to the identification of promising compounds, showcasing a systematic approach to drug discovery.

• Relevance to Public Health: The study addresses a critical need for new cholera treatments, which could have significant implications for global health, particularly in cholera-endemic regions.

Limitations:

1. The study is limited to in vitro models, specifically yeast and HT29 cell lines. While these models are valuable for preliminary screening, the lack of in vivo data is a significant limitation. Animal studies are necessary to confirm the efficacy and safety of novobiocin and apomorphine in living organisms.

Ans: We are in total agreement with the reviewer’s concern. Animal experiments will be conducted in collaboration with proper approval. In an animal model, the effect of apomorphine and novobiocin will also be examined in combination with new strategies. Recently, probiotic-mediated control of Vibrio cholerae has been shown quite effective in animal models (Sengupta et al., 2017; Nag et al., 2018). Similarly, L-ascorbic acid is also able to restrict the growth of Vibrio cholerae under in vitro conditions (Sen et al., 2024). So apomorphine and novobiocin will be examined either with the probiotic or in the presence of L-ascorbic acid. This demands extensive animal experimentation, and therefore, it will be taken up separately. Regarding safety, these are FDA-approved drugs. Apomorphine is still in use to treat Parkinson’s disease. It is interesting to note that some Parkinson’s drugs cause alteration of gut microbiota (van Kessel et al., 2022). This will also be checked in animal and fecal slurry models (Heer et al., 2024). 

References:

1. Sengupta C, Ekka M, Arora S, Dhaware PD, Chowdhury R, Raychaudhuri S.Cross feeding of glucose metabolism byproducts of Escherichia coli human gut isolates and probiotic strains affect survival of Vibrio cholerae. Gut Pathog. 2017 Jan 17;9:3. doi: 10.1186/s13099-016-0153-x. eCollection 2017. 

2. Nag D, Breen P, Raychaudhuri S, Withey JH. Glucose metabolism by Escherichia coli inhibits Vibrio cholerae intestinal colonization of zebrafish. Infect Immun. 2018 Sep 24. pii: IAI.00486-18. Dec; 86(12): e00486-18. doi: 10.1128/IAI.00486-18.

3. Sen, H, Kaur, M, Raychaudhuri S. L-Ascorbic acid restricts Vibrio cholerae survival in various growth conditions. Microorganisms 2024, 12, 492. https://doi.org/10.3390/microorganisms12030492

4. van Kessel SP, Bullock A, van Dijk G, El Aidy S. Parkinson's Disease Medication Alters Small Intestinal Motility and Microbiota Composition in Healthy Rats. mSystems. 2022 Feb 22;7(1):e0119121. doi: 10.1128/msystems.01191-21.

5. Heer K, Kaur, M, Sidhu, D, Dey, P, Raychaudhuri, S. Modulation of gut microbiome in response to the combination of Escherichia coli Nissle 1917 and sugars: a pilot study using host-free system reflecting impact on interpersonal microbiome. Frontiers in Nutrition 2024, 11, 1-17. doi: 10.3389/fnut.2024.1452784

2. The discussion section should include a comparison of the efficacy and potential advantages of novobiocin and apomorphine over existing cholera treatments and other emerging therapeutic approaches. This context would provide a clearer picture of the significance of the findings.

Ans: The discussion section has been modified. 

3. Statistical significance analysis must be performed for all the graphs in the figures.

Ans: We reanalyzed the data using different tools, but the significance of FACS analysis of percentage live cells is low. Kindly see. 

The manuscript is recommended for publication after addressing the limitations listed above.

Reviewer #2: Dear authors,

It is a good work that i think it contribute to the biological and toxicological sciences. I have one comment to improve your work as follows:

Please, prepare a more rigor and professional abstract showcasing your methodologies and results, and the contributions to the science.

Ans: We have modified the sections as suggested. 

Reviewer #3: The resolution of figures seem very low. I could not figure out some of the labeling. I would increase the resolution and size accordingly. I was unable to verify the first two diagrams because of this issue.

Ans: We have used the PLOS journals software for better resolution. Kindly see the new figures. 

6. PLOS authors have the option to publish the peer review history of their article (what does this mean?). If published, this will include your full peer review and any attached files.

Do you want your identity to be public for this peer review? For information about this choice, including consent withdrawal, please see our Privacy Policy.

Reviewer #1: No

Reviewer #2: Yes: Mohammad Amrollahi-Sharifabadi, Ph.D.

Reviewer #3: No

---

## [Decision Letter · Decision Letter 1]

21 Nov 2024

Administration of novobiocin and apomorphine mitigates cholera toxin mediated cellular toxicity: Lessons from cholera toxin yeast model system

PONE-D-24-21529R1

Dear Dr. Raychaudhuri,

We’re pleased to inform you that your manuscript has been judged scientifically suitable for publication and will be formally accepted for publication once it meets all outstanding technical requirements.

Kind regards,

Rui Tada, Ph.D.

Academic Editor

PLOS ONE

Additional Editor Comments (optional):

Reviewers' comments:

Reviewer's Responses to Questions

**Comments to the Author**

1. If the authors have adequately addressed your comments raised in a previous round of review and you feel that this manuscript is now acceptable for publication, you may indicate that here to bypass the “Comments to the Author” section, enter your conflict of interest statement in the “Confidential to Editor” section, and submit your "Accept" recommendation.

Reviewer #1: All comments have been addressed

Reviewer #2: All comments have been addressed

2. Is the manuscript technically sound, and do the data support the conclusions?

Reviewer #1: Yes

Reviewer #2: Yes

3. Has the statistical analysis been performed appropriately and rigorously? 

Reviewer #1: Yes

Reviewer #2: Yes

4. Have the authors made all data underlying the findings in their manuscript fully available?

Reviewer #1: Yes

Reviewer #2: Yes

5. Is the manuscript presented in an intelligible fashion and written in standard English?

Reviewer #1: Yes

Reviewer #2: Yes

6. Review Comments to the Author

Reviewer #1: (No Response)

Reviewer #2: There is no further comments. I think this contribution worth to be published in your journal. Thanks for the opportunity to read your work.

7. PLOS authors have the option to publish the peer review history of their article (what does this mean?). If published, this will include your full peer review and any attached files.

Reviewer #1: No

Reviewer #2: **Yes: **Mohammad Amrollahi-Sharifabadi, PhD in Toxicology

---

## [Editor Report · Acceptance letter]

26 Nov 2024

PONE-D-24-21529R1 

PLOS ONE

Dear Dr. Raychaudhuri, 

I'm pleased to inform you that your manuscript has been deemed suitable for publication in PLOS ONE. Congratulations! Your manuscript is now being handed over to our production team.

Kind regards, 

on behalf of

Dr. Rui Tada 

Academic Editor

PLOS ONE